# Plasma Proteomics in Healthy Subjects with Differences in Tissue Glucocorticoid Sensitivity Identifies A Novel Proteomic Signature

**DOI:** 10.3390/biomedicines10010184

**Published:** 2022-01-16

**Authors:** Nicolas C. Nicolaides, Manousos Makridakis, Rafael Stroggilos, Vasiliki Lygirou, Eleni Koniari, Ifigeneia Papageorgiou, Amalia Sertedaki, Jerome Zoidakis, Evangelia Charmandari

**Affiliations:** 1First Department of Pediatrics, Division of Endocrinology, Metabolism and Diabetes, Aghia Sophia Children’s Hospital, National and Kapodistrian University of Athens Medical School, 11527 Athens, Greece; helenia8@yahoo.it (E.K.); ifipap88@gmail.com (I.P.); aserted@med.uoa.gr (A.S.); evangelia.charmandari@googlemail.com (E.C.); 2Center of Clinical, Experimental Surgery and Translational Research, Division of Endocrinology and Metabolism, Biomedical Research Foundation of the Academy of Athens, 11527 Athens, Greece; 3University Research Institute of Maternal and Child Health and Precision Medicine, 11527 Athens, Greece; 4Department of Molecular Genetics, Function and Therapy, The Cyprus Institute of Neurology and Genetics, 2371 Nicosia, Cyprus; 5Biotechnology Division, Biomedical Research Foundation of the Academy of Athens, 11527 Athens, Greece; mmakrid@bioacademy.gr (M.M.); rstrog@bioacademy.gr (R.S.); vlygirou@bioacademy.gr (V.L.); izoidakis@bioacademy.gr (J.Z.)

**Keywords:** glucocorticoids, glucocorticoid receptor, proteomics, tissue glucocorticoid sensitivity

## Abstract

Significant inter-individual variation in terms of susceptibility to several stress-related disorders, such as myocardial infarction and Alzheimer’s disease, and therapeutic response has been observed among healthy subjects. The molecular features responsible for this phenomenon have not been fully elucidated. Proteomics, in association with bioinformatics analysis, offer a comprehensive description of molecular phenotypes with clear links to human disease pathophysiology. The aim of this study was to conduct a comparative plasma proteomics analysis of glucocorticoid resistant and glucocorticoid sensitive healthy subjects and provide clues of the underlying physiological differences. For this purpose, 101 healthy volunteers were given a very low dose (0.25 mg) of dexamethasone at midnight, and were stratified into the 10% most glucocorticoid sensitive (S) (*n* = 11) and 10% most glucocorticoid resistant (R) (*n* = 11) according to the 08:00 h serum cortisol concentrations determined the following morning. One month following the very-low dose dexamethasone suppression test, DNA and plasma samples were collected from the 22 selected individuals. Sequencing analysis did not reveal any genetic defects in the human glucocorticoid receptor (*NR3C1*) gene. To investigate the proteomic profile of plasma samples, we used Liquid Chromatography–Mass Spectrometry (LC-MS/MS) and found 110 up-regulated and 66 down-regulated proteins in the S compared to the R group. The majority of the up-regulated proteins in the S group were implicated in platelet activation. To predict response to cortisol prior to administration, a random forest classifier was developed by using the proteomics data in order to distinguish S from R individuals. Apolipoprotein A4 (APOA4) and gelsolin (GSN) were the most important variables in the classification, and warrant further investigation. Our results indicate that a proteomics signature may differentiate the S from the R healthy subjects, and may be useful in clinical practice. In addition, it may provide clues of the underlying molecular mechanisms of the chronic stress-related diseases, including myocardial infarction and Alzheimer’s disease.

## 1. Introduction

Glucocorticoids exert numerous actions in target tissues through binding to their cognate receptor, the glucocorticoid receptor (GR), which functions as a ligand-activated transcription factor influencing the transcription rate of a large number of glucocorticoid responsive genes [1,2,3]. Moreover, the ligand-bound GR may influence gene expression in a positive or negative fashion through protein–protein interactions with other transcription factors, including the nuclear factor-κB (NF-κB) [4], signal transducers and activators of transcription (STATs) [5], and the activator protein-1 (AP-1) [4,6]. In addition to their genomic actions, glucocorticoids may exert rapid nongenomic actions mediated by a membrane-anchored GR, which triggers the activation of the mitogen-activated protein kinases (MAPK) [7,8] or the phosphatidylinositol 3-kinase (PI_3_K) [8,9]. Finally, accumulating evidence suggests that glucocorticoids may signal through mitochondrial GRs, thereby regulating fundamental cellular function, including apoptosis, energy production, and stress response [10].

In humans, tissue glucocorticoid sensitivity is complex and multifactorial, depending on polymorphisms [11] or genetic defects in the *NR3C1* gene that encodes the human GR (hGR) [12], several tissue-specific hGR protein isoforms [13,14], or the hGR “interactome” influenced by an ever-increasing number of interacting proteins, miRNAs [15] and non-coding RNAs [15,16]. All these regulatory factors influence substantially tissue responsiveness to glucocorticoids in clinical practice. Indeed, a large number of patients with inflammatory, allergic, hematologic, and lymphoproliferative diseases are treated with synthetic glucocorticoids due to their anti-inflammatory and immunosuppressive effects [17]. However, the therapeutic outcome might be compromised by the adverse effects of hypercortisolism [18]. Therefore, the identification of novel protein biomarkers that could stratify healthy adults as most glucocorticoid-sensitive or most glucocorticoid-resistant is currently an unmet clinical need.

Proteomics approaches allow the identification and quantification of proteins from biological samples and could lead to the identification of biomarkers associated with the pathogenesis, diagnosis, and treatment of many diseases [19]. Of note, untargeted proteomics has not yet been used to investigate differences among healthy individuals with variations in tissue glucocorticoid sensitivity. The aim of our study was to study any differences at the proteomics level among healthy participants with increased or decreased sensitivity to glucocorticoids, and to identify a proteomic signature that could be used to distinguish most glucocorticoid-sensitive (S) from most glucocorticoid-resistant (R) healthy adult subjects, who do not harbor any genetic defects in the human glucocorticoid receptor (*NR3C1*) gene, using untargeted plasma proteomics.

## 2. Materials and Methods

### 2.1. Selection of the Healthy Subject Cohort

Fifty males and fifty-one females (*n* = 101) of a mean age of (±SD) 26.5 (±5.0) years, were recruited to participate in our study prospectively. The participants had unremarkable medical history and were not receiving any medications, including oral contraceptives for females. To study the inter-individual variation in tissue glucocorticoid sensitivity among the participants, the very low dose dexamethasone suppression test was applied, as previously described in studies with healthy subjects [20,21]. According to this protocol, all participants were given per os 0.25 mg of dexamethasone at midnight and were advised to wake up at 07.00 h. The following morning (08.00 h), serum cortisol (261.8 ± 206.9 nmol/L) and plasma ACTH (16.1 ± 12.2 pg/mL) were measured. Depending on their serum cortisol concentrations, all participants were rank ordered. The 11 subjects who had the lowest serum cortisol concentrations were classified as the most glucocorticoid sensitive (S) group, while the 11 subjects with the highest serum cortisol concentrations were classified as the most glucocorticoid-resistant (R) group ((mean serum cortisol concentrations ± SD: 34.4 ± 15.0 nmol/L in the S participants, mean serum cortisol concentrations ± SD: 622.4 ± 93.7 nmol/L in the R participants, *p* < 0.001); (mean plasma ACTH concentrations ± SD: 2.8 ± 2.4 pg/mL in the S participants, mean plasma ACTH concentrations ± SD: 31.6 ± 10.6 pg/mL in the R participants, *p* < 0.001)) (Table 1). Thirty days following the very low dose dexamethasone suppression test, plasma, DNA and RNA samples were collected from the 22 individuals for further analyses.

### 2.2. Approval of the Study

The study was approved by the Aghia Sophia Children’s Hospital Committee on the Ethics of Human Research EB-PASCH-MoM: 13/02/2013, Re: 1490-21/01/2013). Written informed consent was obtained by all subjects prior to their participation in the study.

### 2.3. Sample Collection

Whole blood samples (total volume: 5 mL) were obtained from the study participants at 08:00 h in EDTA-containing tubes, centrifuged immediately following collection, and stored at −80 °C until further analysis. Two aliquots of 150 μL per blood sample were shipped in dry ice to the proteomics laboratory.

### 2.4. Assays

The standard hematological, biochemical, and endocrinological investigations were determined, as previously described [22]. Taking into account the reproductive cycle of the females, gonadotropin and sex steroid concentrations were determined between the 3rd and 5th day of the menstrual cycle.

### 2.5. Sequencing of the NR3C1 Gene

Genomic DNA was isolated from whole blood samples from the 22 selected participants using the Maxwell 16 instrument for automated DNA extraction (Promega Corp., Madison, WI, USA). The protein-coding sequences of the *NR3C1* gene, including the junctions of introns and exons, were PCR-amplified and sequenced using the Big Dye Terminator cycle sequencing kit (Applied Biosystems, Carlsbad, CA, USA) on an ABI 3100 sequencer (ABI 3100, Applied Biosystems, Carlsbad, CA, USA), as previously described [22,23].

### 2.6. Sample Preparation for Proteomics Analysis

Plasma samples (200 μg total protein per sample) were processed with the filter aided sample preparation (FASP) method as described previously [24], with minor modifications [25]. Briefly, proteins were reduced with DTE (0.1 M), alkylated with iodoacetamide (0.05 M), and digested overnight by trypsin in 50 mM NH_4_HCO_3_ pH 8.5. The peptides were lyophilized and kept at −80 °C.

### 2.7. LC-MS/MS Analysis

Samples were resuspended in 200 μL mobile phase A (0.1% FA). A 5 μL volume was injected into a Dionex Ultimate 3000 RSLS nano flow system (Dionex, Camberly, UK) configured with a Dionex 0.1 × 20 mm, 5 μm, 100 Å C18 nano trap column with a flow rate of 5 µL/min. The analytical column was an Acclaim PepMap C18 nano column 75 μm × 50 cm, 2 μm 100 Å with a flow rate of 300 nL/min. The trap and analytical column were maintained at 35 °C. The chromatography method used had a duration of 4 h and peptide elution was achieved by the following gradient: 2% to 33% mobile phase B (ACN 100%, Formic acid 0.1%). MS/MS analysis was performed with a Q Exactive operated with Data Dependent Acquisition mode.

### 2.8. MS Data Processing

Proteome Discoverer 1.4 was used for processing the raw mass spectrometry data with the Sequest search algorithm and the Uniprot FASTA file for *Homo sapiens* (20,243 entries downloaded on 15 December 2017). Cysteine cabamidomethylation was considered as a fixed modification, whereas methionine oxidation was considered as variable modification. The mass tolerance for precursors was set to 10 ppm, whereas for fragment ions it was 0.05 Da. The False discovery rate (FDR) was set to 0.01.

### 2.9. Data Processing (Methods)

Data processing, visualizations, and statistical analysis were conducted in the R environment (version 3.6) for Windows. Raw spectral areas per sample were merged with an in-house script and were subjected for sample normalization according to X′ = [X/sum(Xi)] × 10^6^, with X being the raw area of a given protein ID for a given sample, sum(Xi) the sum of all raw protein areas for the same sample and X′ the normalized protein area. To increase the power of the downstream analysis, proteins showing null abundance in more than 65% of samples in both biological groups (“hypersensitive” and “resistant”; described in the results section) were excluded. The non-parametric Mann–Whitney test was utilized for assessing statistical significance of the continuous variables. Heatmap was constructed after row z-normalization with the package ComplexHeatmap and the volcano plot with the package EnhancedVolcano. Pathway analysis was conducted in the Cytoscape plug-in ClueGO+CluePedia [26] and significantly enriched pathways were defined with a two-sided hypergeometric test corrected with Benjamini–Hochberg (*p* < 0.05). Grouping of the enriched terms was based on 50% gene similarity. A random forest-based classifier was trained to distinguish between the two biological groups using the package randomForest. The complete study population was utilized for training and feature selection involved proteins with no missing values across samples and specifically the union of statistically significant proteins (defined by Mann–Whitney *p* < 0.05; *n* = 11) and of proteins with fold change >1.5 (*n* = 3 extra proteins). Tuning of classifier parameters was conducted iteratively and was based on minimizing the out of bag error [27], which is an estimate of the correct classification rate of random partitions of the training set. For optimal tuning, number of trees and variables used at each split was set to 48 and 3, respectively, with a stepfactor of 0.5 and improve of 0.05. Variable importance was assessed in terms of the increase in the misclassification rate after permutation and comparison of the accuracy to the standard model (Mean Decrease in Accuracy and Mean Decrease in Gini index; [28]) and also in terms of the mean minimal depth achieved over all trees, which indicates the impact of a variable in the classification of observations (the lower the minimal depth of a variable the highest the number of observations that were classified on the basis of that variable), with the packages caret and randomForestExplainer.

Summaries and detailed data on protein intensities and statistics can be found in the Appendix A.

## 3. Results

### 3.1. Clinical Characteristics, Hematological, Biochemical, and Endocrinological Parameters in the Most Glucocorticoid Sensitive (S) and Most Glucocorticoid Resistant (R) Healthy Subjects

The clinical characteristics (sex, weight, height, and body mass index (BMI)), serum cortisol and plasma ACTH concentrations at the time following the very low dose dexamethasone suppression test are presented in Table 1. All the hematological, biochemical and endocrinological parameters of the S and R healthy subjects measured one month after the very-low dexamethasone suppression test are presented in Table 2. No statistically significant differences were found between the S and R groups (*p*-value > 0.05).

### 3.2. The S and R Healthy Participants Did Not Harbor Any Genetic Defects in the NR3C1 Gene

To study whether the observed variation in tissue glucocorticoid sensitivity among the S and R healthy participants might be caused by genetic defects in the *NR3C1* gene, the protein-coding exons and the intron–exon junctions were first PCR amplified and then sequenced. We have not identified any point mutations, deletions, insertions, or polymorphisms in the *NR3C1* gene of the S and R participants.

### 3.3. Proteomics Data Analysis

In total, there were 2737 proteins identified and quantified in at least one of the analyzed samples (listed in the Appendix A). Based on their response to cortisol, samples were labeled either as “most sensitive” (mean serum cortisol concentrations ± SD: 30 ± 20 nmol/L; *n* = 11 samples) or “most resistant” (mean serum cortisol concentrations ± SD: 620 ± 90 nmol/L; *n* = 11 samples), and statistical analysis was conducted for the comparison of resistant vs sensitive. After selecting those proteins with presence in at least 35% of the samples in one of the two groups (*n* = 466 features), differentially expressed proteins were defined as the subset of statistically significant changes (Mann–Whitney *p*-value < 0.05) with a fold change greater than 1.5 (or less than 0.67). This counted for 66 proteins with higher abundance in the resistant and 110 proteins with higher abundance in the sensitive group (Figure 1). Among them, there were 21 proteins being present exclusively in only one of the two groups (Table 3). Over-representation test (ClueGO plug-in) for proteins being overexpressed in the sensitive group (FC < 0.67; *n* = 110 proteins) yielded some significantly enriched Reactome pathways (Table 4), including erythrocyte gas exchange and platelet activation and aggregation. However, there were no significant pathway enrichments for proteins being at higher abundance in the resistant group (FC > 1.5; *n* = 66 proteins). Detailed protein data for this analysis can be found in the “35% threshold” spreadsheet of the Appendix A.

In order to predict the response to cortisol prior to administration, a random forest classifier was developed out of the proteomics data, to distinguish between sensitive and resistant samples. As a training set, we used the complete set of samples (*n* = 22) aiming to validate the classifier in new independent cohorts. For optimal training, feature selection involved proteins with no missing values across samples and specifically those passing Mann–Whitney *p* < 0.05, as well as those with a fold change >1.5 (and <0.67; *n* = 14). After tuning for optimal parameters, the classifier showed promising results in correctly assigning random partitions of the training data to the studied groups, achieving an overall accuracy score of 0.86. In detail, there were only three misclassified cases, one for the sensitive and two for the resistant groups. The individual importance of each protein in the model was evaluated in terms of the Mean Decrease in Accuracy, the Mean Decrease in Gini index and the mean minimal depth. Out of the 14 proteins utilized for training, apolipoprotein A4 (APOA4) and gelsolin (GSN) were the most important variables in the classification (Figure 2), and warrant further investigation to determine their prognostic capacity.

## 4. Discussion and Conclusions

In this study, 22 individuals were selected as the most glucocorticoid sensitive (S) (*n* = 11) and most glucocorticoid resistant (R) (*n* = 11) from a large sample of 101 healthy adults, according to their 08.00 h serum cortisol concentrations following the very low dose dexamethasone suppression test. This test has been widely used in studies aimed to identify mild differences in tissue glucocorticoid sensitivity due to interindividual variation among healthy adults [20,21]. Since serum cortisol concentrations are within normal range in these studies, the application of a cortisol cut-off was out of context; therefore, the 101 participants were rank-ordered according to their cortisol concentrations and the two normal “extreme” groups were selected for further analyses. Sequencing analysis of the *NR3C1* gene did not identify any genetic defect in the 22 participants, indicating that the above-mentioned differences at the proteomics level could not be attributed to increased or decreased expression or activity of the human glucocorticoid receptor.

Untargeted plasma proteomics analysis revealed 110 up-regulated and 66 down-regulated proteins in the S compared to the R group. Importantly, the majority of the up-regulated proteins in the S group were implicated in erythrocyte gas exchange (take up carbon dioxide and release oxygen), suggesting a state of the organism that is more capable to respond to stressful stimuli. This result is in line with the basic concepts of physiology of the stress system [29,30,31,32]. Upon exposure to external or internal stressors, a highly conserved neuroendocrine system, the stress system, is activated to restore the threatened or perceived as threatened internal balance or homeostasis. To achieve homeostasis, the stress system, through concurrent activation of the hypothalamic-pituitary–adrenal (HPA) axis and the locus caeruleus–autonomic nervous system, leads to behavioral and physical adaptation changes, thereby increasing the chances for survival [29,30,31,32]. Thus, both oxygen and nutrients are redirected to the central nervous system and the stressed body site(s) through increased respiratory rate, elevated cardiovascular tone and catabolism [29,30]. The S group had most of the up-regulated proteins involved in erythrocyte take up of carbon dioxide and release of oxygen, probably as part of the physical adaptation changes observed in stress-related homeostasis.

In addition to the increased gas exchange of erythrocytes, the S group was characterized by increased expression of proteins involved in platelet activation, aggregation, and degranulation. Of note, there were not any statistically significant differences between the number of platelets of the S participants compared to that of the R participants, indicating that the S group displayed a cellular “predisposition” for increased coagulation. A growing body of evidence suggests that natural and synthetic glucocorticoids contribute substantially to platelet activation and aggregation [33]. Indeed, the serum/glucocorticoid regulated kinase 1 (SGK1), which is a regulator of ORAI1 protein that mediates the store-operated Ca^2+^ entry in platelets, has been shown to be influenced by several hormones, including glucocorticoids [34]. Furthermore, endogenous hypercortisolism has been found to increase thromboxane A2 (TXA2) biosynthesis, leading to enhanced platelet aggregation [35]. The increased platelet activity and aggregation underlies the increased coagulation causing blood vessel infarctions, including myocardial and/or brain infarctions, often observed in chronically stressed individuals. Although the S participants did not have endogenous or exogenous hypercortisolism, we speculate that their increased stress response might predispose them to develop blood vessel infarction upon exposure to chronic stress.

Our study also succeeded in identifying a protein signature that could distinguish the S from the R participants. Out of the 14 proteins utilized for training in the random forest classifier, APOA4 and gelsolin GSN were the most important variables in the classification. The expression of ApoA4 was significantly higher in obese and overweight children compared with their normal counterparts. ApoA4 has an antiatherogenic function [36], is increased in obese subjects and declines with weight reduction [37]. Indeed, fasting plasma ApoA4 concentrations are significantly elevated in obesity and decrease to almost half of baseline concentrations during weight reduction [37]. In addition to APOA4, GSN was also found as one of the most important variables in the classification. GSN is one of the most abundant actin-binding proteins and plays a crucial role in cellular mechanisms and interactions [38]. Since GSN participates in several immunologic functions, and interacts with different types of immune cells, this protein has been identified as a potential therapeutic target. More specifically, previous studies have identified the anti-amyloidogenic role of GSN in Alzheimer’s disease (AD) [39,40,41]. GSN can reduce amyloid burden by acting as an inhibitor of amyloid beta-protein (Aβ) fibrillization, and as an antioxidant and anti-apoptotic protein. When GSN was administered or overexpressed in AD transgenic mice, the amyloid load was significantly reduced and the Aβ level was decreased, suggesting that GSN might be implicated in the treatment of AD [39]. The anti-amyloidogenic activity in the S participants is in line with the results of our previous published study, in which we used transcriptomics in these 22 participants, and found that the S participants had decreased expression of genes involved in Alzheimer’s disease [23].

To the best of our knowledge, this is the first pilot study that identified differences at the proteomics level among healthy adults with variations in tissue glucocorticoid sensitivity. However, the study has some limitations. The sample of participants was relatively small to extract concrete conclusions. Undoubtedly, the proteomics analysis must be repeated in a larger cohort in order to confirm the differences reported between S and R individuals. In the future, it is planned to validate the differential expression of selected proteins and the deregulation of key pathways (erythrocyte gas exchange, and platelet activation) by independent analytical approaches (ELISA, flow cytometry, etc.). Another limitation of our study was the use of IMMULITE instead of HPLC for measuring serum cortisol concentrations. The use of IMMULITE is the standard procedure that we normally follow in our every-day clinical practice. Finally, the dexamethasone concentrations of the 101 participants were not measured following the very-low dexamethasone suppression test because we carried out the protocol performed by Donn and collaborators, who found a new glucocorticoid sensitivity-determining gene using gene expression profiling [21]. It is worth mentioning that in most published studies that use the very-low dose dexamethasone suppression test, the investigators have not routinely determined dexamethasone concentrations in their participants.

In conclusion, untargeted proteomics analysis revealed differences among healthy subjects with differences in glucocorticoid sensitivity. The S individuals displayed higher erythrocyte gas exchange and increased platelet activation and aggregation that might lead to increased risk of stress-related disorders, including myocardial and brain infarctions.

## Figures and Tables

**Figure 1 biomedicines-10-00184-f001:**
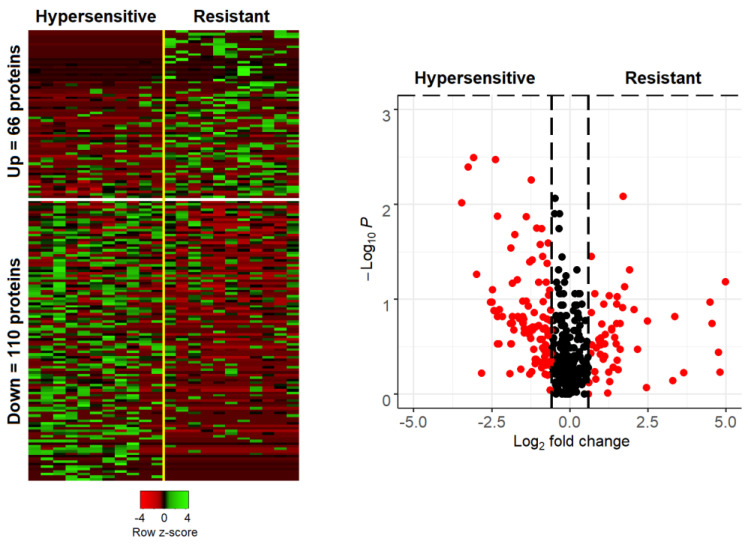
Heatmap (**left**) and Volcano plot (**right**) of proteins quantified in patients with complete (hypersensitive) or not complete (resistant) response to cortisol. Heatmap shows the abundance of proteins passing the ±0.585 log2 fold change threshold, in the two groups. Volcano plot includes proteins with presence in at least 35% of the analyzed samples (in one of the two groups) and illustrates the log2 fold change (x axis) as a function of the Mann–Whitney p value (y axis). Red color marks proteins passing the 1.5 (or 0.67)-fold change (equivalent to ±0.585 in the logarithmic scale).

**Figure 2 biomedicines-10-00184-f002:**
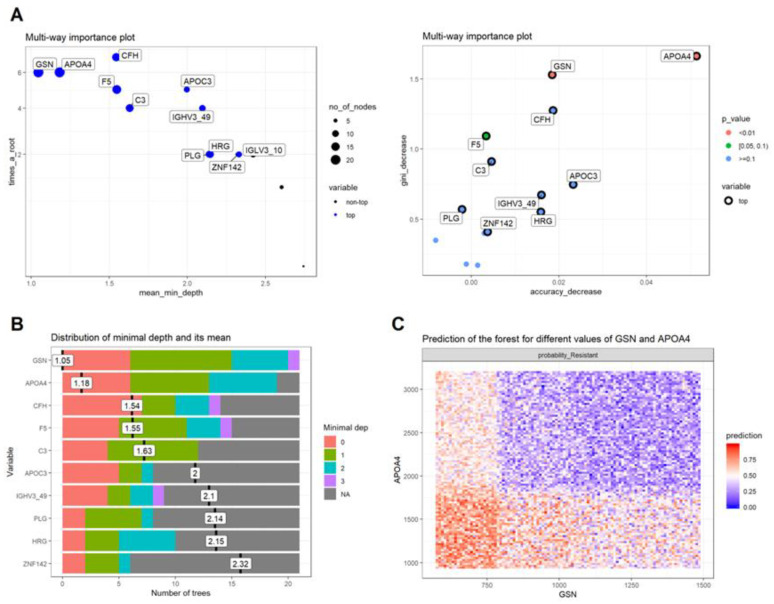
Variable importance for the 14 proteins used to train the random forest classifier so as to distinguish between responders (hypersensitive) and non-responders (resistant) to cortisol: (**A**) Multiway importance plots depicting the mean decrease in accuracy as a function of the mean minimal depth (left) and of the mean decrease in the Gini index (right); (**B**) Plot showing structure of the forest with respect to the distribution of the mean minimal depth across trees for each variable; (**C**) Predictive value of the top two important variables (APOA4 and GSN) in detecting non-responders, based on their normalized protein intensity areas across samples.

**Table 1 biomedicines-10-00184-t001:** Clinical characteristics, and serum cortisol and plasma ACTH concentrations of the most glucocorticoid sensitive (S) and most glucocorticoid resistant (R) healthy subjects at the time of the very-low dexamethasone suppression test.

Group	Sample Code	Sex	Weight (kg)	Height (m)	BMI (kg/m^2^)	Cortisol (nmol/L)	ACTH (pg/mL)
Glucocorticoid Sensitive (S)	1	F	58	1.64	21.6	18.6	<1.0
	2	F	62	1.75	20.2	22.2	1.4
	3	M	70	1.77	22.3	23.1	6.2
	4	F	45	1.50	20,0	24.5	<1.0
	5	M	70	1.85	20.5	26.2	2.9
	6	F	55	1.64	20.4	32.3	<1.0
	7	F	48	1.57	19.5	34.2	5.1
	8	M	80	1.78	25.2	36.1	<1.0
	9	M	70	1.82	21.1	39.7	2.0
	10	M	52	1.71	17.8	51.3	<1.0
	11	M	81	1.87	23.2	69.5	7.6
Glucocorticoid Resistant (R)	1	F	52	1.59	20.6	834.0	35.3
	2	F	56	1.68	19.8	720.9	38.1
	3	F	59	1.55	24.6	690.8	46.0
	4	M	93	1.86	26.9	644.2	42.2
	5	M	53	1.68	18.8	599.0	32.8
	6	F	47	1.54	19.8	597.9	23.7
	7	F	59	1.70	20.4	579.4	39.9
	8	F	58	1.65	21.3	565.3	16.1
	9	F	58	1.7	20.1	556.2	29.9
	10	M	70	1.72	23.7	537.4	30.9
	11	M	77	1.88	21.8	520.6	12.4

BMI: Body Mass Index; ACTH: Adrenocorticotropic Hormone.

**Table 2 biomedicines-10-00184-t002:** Hematological, biochemical, and endocrinological parameters of the S and R healthy subjects determined one month after the very-low dexamethasone suppression test.

Parameter	Glucocorticoid Sensitive (*n* = 11)	Glucocorticoid Resistant (*n* = 11)	** p*
Age (year)	25.27 ± 1.17	27.55 ± 2.03	0.478
Weight (kg)	62.82 ± 3.72	62.00 ± 3.98	0.847
Height (cm)	1.72 ± 0.04	1.69 ± 0.03	0.519
BMI (kg/m^2^)	21.07 ± 0.60	21.62 ± 0.74	0.797
ACTH (pg/mL)	33.16 ± 5.67	27.64 ± 4.65	0.519
CORT (μg/dL)	23.13 ± 1.70	18.98 ± 3.06	0.270
IGF-I (ng/mL)	259.18 ± 23.97	251.36 ± 20.15	0.699
IGFBP-3 (μg/mL)	5.30 ± 0.31	5.17 ± 0.37	0.562
TSH (μUI/mL)	2.79 ± 0.28	2.05 ± 0.33	0.101
T3 (ng/dL)	102.26 ± 8.33	102.02 ± 7.19	0.982
FT4 (ng/dL)	1.12 ± 0.04	1.06 ± 0.03	0.261
Anti-TPO (IU/mL)	10.43 ± 0.21	11.11 ± 0.78	0.652
Anti-TG (IU/mL)	20.00 ± 0.00	20.00 ± 0.00	0.999
LH (mUI/mL)	10.11 ± 4.50	6.44 ± 0.69	0.699
FSH (mUI/mL)	5.22 ± 0.83	4.05 ± 0.70	0.300
DHEAS (μg/dL)	238.62 ± 44.03	248.58 ± 34.68	0.562
Androstenedione (ng/mL)	2.89 ± 0.28	3.20 ± 0.35	0.502
PRL (ng/mL)	24.94 ± 2.65	21.55 ± 2.75	0.193
SHBG (nmol/L)	65.12 ± 8.42	46.17 ± 5.09	0.175
PTH (pg/mL)	34.15 ± 4.59	38.51 ± 5.40	0.562
25-Hydroxy-Vitamin D (ng/mL)	16.06 ± 2.38	14.02 ± 2.56	0.652
Insulin (μUI/mL)	6.71 ± 0.81	13.72 ± 4.22	0.116
Glucose (mg/dL)	73.20 ± 1.99	74.75 ± 4.80	0.965
Urea (mg/dL)	27.70 ± 2.06	32.50 ± 2.90	0.203
Cholesterol (mg/dL)	157.40 ± 5.34	156.75 ± 5.30	0.965
HDL (mg/dL)	49.50 ± 2.21	52.88 ± 2.86	0.315
LDL (mg/dL)	90.70 ± 5.63	87.63 ± 4.78	0.762
Triglycerides (mg/dL)	69.40 ± 9.48	74.25 ± 5.66	0.315
ApoA1 (mg/dL)	158.40 ± 2.54	167.63 ± 5.32	0.237
ApoB (mg/dL)	75.50 ± 4.55	71.38 ± 2.73	0.515
Lpa (mg/dL)	21.84 ± 11.82	25.79 ± 9.66	0.460
Hct (%)	43.09 ± 1.07	44.64 ± 2.32	0.748
WBC (×10^3^/μL)	6.72 ± 0.49	7.09 ± 0.52	0.612
PLT (×10^3^/μL)	236.55 ± 22.11	233.09 ± 32.56	0.847

ACTH: Adrenocorticotropic Hormone; Anti-Tg: Thyroglobulin antibodies; Anti-TPO: Thyroid Peroxidase antibodies; ApoA1: Apolipoprotein A1; ApoB: Apolipoprotein B; ΒΜΙ: Body Mass Index; CORT: Cortisol; DHEAS: Dehydroepiandrosterone Sulfate; FSH: Follicle Stimulating Hormone; FT4: Free Thyroxine; Hct: Hematocrit; HDL: High-Density Lipoprotein; IGF1: Insulin Like Growth Factor 1; IGF-BP3: Insulin Like Growth Factor-Binding Protein 3; INS: Insulin; LDL: Low-Density Lipoprotein; LH: Luteinizing Hormone; Lpa: lipoprotein a; PLT: Platelet count; PRL: Prolactin; PTH: Parathormone; SHBG: Sex Hormone-Binding Globulin; T3: Triiodothyronine; TSH: Thyroid Stimulating Hormone; WBC: White Blood Cell Count. Data are presented as mean ± standard error of the mean (SEM); * *p* > 0.05 for all.

**Table 3 biomedicines-10-00184-t003:** Proteins being exclusively present in one of the two groups.

Protein	Description	*p* Value	Present
KIF28P	Kinesin-like protein KIF28P	0.015762	Only in resistant
MRPS34	28S ribosomal protein S34, mitochondrial	0.015762	Only in resistant
PRPF8	Pre-mRNA-processing-splicing factor 8	0.015762	Only in resistant
MYH11	Myosin-11	0.03591	Only in resistant
MLH1	DNA mismatch repair protein Mlh1	0.03591	Only in resistant
ARHGAP21	Rho GTPase-activating protein 21	0.03591	Only in resistant
EMC10	ER membrane protein complex subunit 10	0.03591	Only in resistant
ZSWIM9	Uncharacterized protein ZSWIM9	0.03591	Only in resistant
FANCB	Fanconi anemia group B protein	0.03591	Only in resistant
CDADC1	Cytidine and dCMP deaminase domain-containing protein 1	0.03591	Only in resistant
ACSS3	Acyl-CoA synthetase short-chain family member 3, mitochondrial	0.03591	Only in resistant
IGHV3-66	Immunoglobulin heavy variable 3-66	0.03591	Only in sensitive
IGLV5-39	Immunoglobulin lambda variable 5-39	0.03591	Only in sensitive
LCP1	Plastin-2	0.03591	Only in sensitive
DOCK4	Dedicator of cytokinesis protein 4	0.03591	Only in sensitive
SLC38A3	Sodium-coupled neutral amino acid transporter 3	0.03591	Only in sensitive
RTN4	Reticulon-4	0.03591	Only in sensitive
CFAP97	Cilia- and flagella-associated protein 97	0.03591	Only in sensitive
POLK	DNA polymerase kappa	0.03591	Only in sensitive
ANKRD50	Ankyrin repeat domain-containing protein 50	0.03591	Only in sensitive

**Table 4 biomedicines-10-00184-t004:** Pathway enrichments for the sensitive group. Significance was defined with a two-sided hypergeometric test in ClueGo. P value corresponds to the Benjamini–Hochberg correction.

Reactome Pathway	*p* Value	% Associated Genes	Associated Genes Found
Erythrocytes take up oxygen and release carbon dioxide	7.07 × 10^−5^	33.3	(CA1, CA2, HBA1)
G-protein mediated events	0.004559	5.5	(CAMKK2, ITPR1, ITPR2)
PLC beta mediated events	0.004452	5.6	(CAMKK2, ITPR1, ITPR2)
DAG and IP3 signaling	0.002614	7.1	(CAMKK2, ITPR1, ITPR2)
Signaling by VEGF	0.000892	4.7	(CDH5, CRK, ITGB3, ITPR1, ITPR2)
VEGFA-VEGFR2 Pathway	0.0007	5.1	(CDH5, CRK, ITGB3, ITPR1, ITPR2)
Platelet activation, signaling and aggregation	1.5 × 10^−6^	4.6	(CRK, F8, FLNA, ITGB3, ITPR1, ITPR2, PFN1, PPBP, QSOX1, RARRES2, TUBA4A, VCL)
Fcgamma receptor (FCGR) dependent phagocytosis	0.002295	4.7	(CRK, FCGR3A, ITPR1, ITPR2)
Platelet degranulation	1.18 × 10^−6^	7.0	(F8, FLNA, ITGB3, PFN1, PPBP, QSOX1, RARRES2, TUBA4A, VCL)
Response to elevated platelet cytosolic Ca^2+^	1.1 × 10^−6^	6.7	(F8, FLNA, ITGB3, PFN1, PPBP, QSOX1, RARRES2, TUBA4A, VCL)
Role of phospholipids in phagocytosis	0.000858	12.0	(FCGR3A, ITPR1, ITPR2)

## Data Availability

All the data are available in the Appendix A.

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
