# Peer review of "Plasma Proteomics in Healthy Subjects with Differences in Tissue Glucocorticoid Sensitivity Identifies A Novel Proteomic Signature"

_biomedicines, 2022, doi:10.3390/biomedicines10010184_

Round 1

Reviewer 1 Report

The current study reports specific gene signature changes in individual who are sensitive or resistant to glucocorticoids. Overall very well done with detailed description of methodologies and analyses of the data.

Minor concern

In the abstract first line authors mention stress related disorders, can you specifically describe which diseases are referred to, since most of them are stress related.

In materials and methods section all participants received a single dose of dexamethasone (0.25mg) and serum cortisol (261.8 ± 206. 9 nmol/L) and plasma ACTH 16.1± 12.2 pg/ml) were determined. Why such variation, is it a typographical error, or due to variation in the assay? if so a comment from the authors need to make as to why such a large deviations are observed.

Author Response

We thank the Reviewer for his/her comment. We have now added myocardial infarction and Alzheimer’s disease as representative examples of stress-related disorders (page 1, highlighted in yellow color).

We thank the Reviewer for his/her comment. The high variation in the measurements of serum cortisol and plasma ACTH concentrations was observed because the sample of 101 participants included both the sensitive participants with the lowest cortisol and ACTH concentrations, and the resistant participants with the highest cortisol and ACTH concentrations.

Reviewer 2 Report

The paper entitled " Plasma Proteomics in Healthy Subjects with Differences in Tissue Glucocorticoid Sensitivity Identifies a Novel Proteomic Signature " is well structured and presented.

The authors should take in account the wake-up hour of the participants in the low dose dexamethasone suppression test, because it could interfere with the morning cortisol peak and with the selection of the glucocorticoid-sensitive (S) and glucocorticoid-resistant (R) participants.

I also recommend that the authors reflect about the limitations of the study and include some information in order to avoid misunderstandings, taking into account the need of up-regulated and down-regulated proteins validation by other methods. Moreover, the higher erythrocyte gas exchange and increased platelet activation and aggregation was also not validated by other analytical approaches.

Author Response

We agree with the Reviewer. All participants were advised to wake up at 07.00h the following morning (page 3, line 85, highlighted in yellow color).

We thank the Reviewer for this constructive comment. The following statement was added to the discussion in order to highlight the limitations of our study and the future directions of our research efforts (page 16, lines 355-360): “Undoubtedly, the proteomics analysis must be repeated in a larger cohort in order to confirm the differences reported between S and R individuals. In the future, we plan to validate the differential expression of selected proteins and the deregulation of key pathways (erythrocyte gas exchange, platelet activation) by independent analytical approaches (ELISA, flow cytometry, etc.)”

Round 2

Reviewer 2 Report

The revision performed by the authors are adequated.